# NeuMiss networks: differentiable programming for supervised learning with missing values

Marine Le Morvan[1,2]   Julie Josse[1,3]   Thomas Moreau[1]   Erwan Scornet[3]   Gaël Varoquaux[1, 4]

[1] Université Paris-Saclay, Inria, CEA, Palaiseau, 91120, France
[2] Université Paris-Saclay, CNRS/IN2P3, IJCLab, 91405 Orsay, France
[3] CMAP, UMR7641, Ecole Polytechnique, IP Paris, 91128 Palaiseau, France
[4] Mila, McGill University, Montréal, Canada

{marine.le-morvan, julie.josse, thomas.moreau, gael.varoquaux}@inria.fr
erwan.scornet@polytechnique.edu

## Abstract

The presence of missing values makes supervised learning much more challenging. Indeed, previous work has shown that even when the response is a linear function of the complete data, the optimal predictor is a complex function of the observed entries and the missingness indicator. As a result, the computational or sample complexities of consistent approaches depend on the number of missing patterns, which can be exponential in the number of dimensions. In this work, we derive the analytical form of the optimal predictor under a linearity assumption and various missing data mechanisms including Missing at Random (MAR) and self-masking (Missing Not At Random). Based on a Neumann-series approximation of the optimal predictor, we propose a new **principled** architecture, named NeuMiss networks. Their originality and strength come from the use of a new type of non-linearity: the multiplication by the missingness indicator. We provide an upper bound on the Bayes risk of NeuMiss networks, and show that they have good predictive accuracy with both a number of parameters and a computational complexity independent of the number of missing data patterns. As a result they **scale well** to problems with many features, and remain **statistically efficient** for medium-sized samples. Moreover, we show that, contrary to procedures using EM or imputation, they are **robust to the missing data mechanism**, including difficult MNAR settings such as self-masking.

## 1   Introduction

Increasingly complex data-collection pipelines, often assembling multiple sources of information, lead to datasets with incomplete observations and complex missing-values mechanisms. The pervasiveness of missing values has triggered an abundant statistical literature on the subject [14, 31]: a recent survey reviewed more than 150 implementations to handle missing data [10]. Nevertheless, most methods have been developed either for inferential purposes, i.e. to estimate parameters of a probabilistic model of the fully-observed data, or for imputation, completing missing entries as well as possible [6]. These methods often require strong assumptions on the missing-values mechanism, i.e. either the missing at random (MAR) assumption [27] – the probability of being missing only depends on observed values – or the more restrictive Missing Completely At Random assumption (MCAR) – the missingness is independent of the data. In MAR or MCAR settings, good imputation is sufficient to fit statistical models, or even train supervised-learning models [11]. In particular, a precise knowledge

of the data-generating mechanism can be used to derive an Expectation Maximization (EM) [2] formulation with the minimum number of necessary parameters. Yet, as we will see, this is intractable if the number of features is not small, as potentially $2^d$ missing-value patterns must be modeled.

The last missing-value mechanism category, Missing Not At Random (MNAR), covers cases where the probability of being missing depends on the unobserved values. This is a frequent situation in which missingness cannot be ignored in the statistical analysis [12]. Much of the work on MNAR data focuses on problems of identifiability, in both parametric and non-parametric settings [29, 20–22]. In MNAR settings, estimation strategies often require modeling the missing-values mechanism [9]. This complicates the inference task and is often limited to cases with few MNAR variables. Other approaches need the masking matrix to be well approximated with low-rank matrices [18, 1, 7, 16, 32].

Supervised learning with missing values has different goals than probabilistic modeling [11] and has been less studied. As the test set is also expected to have missing entries, optimality on the fully-observed data is no longer a goal per se. Rather, the goal of minimizing an expected risk lend itself well to non-parametric models which can compensate from some oddities introduced by missing values. Indeed, with a powerful learner capable of learning any function, imputation by a constant is Bayes consistent [11]. Yet, the complexity of this function that must be approximated governs the success of this approach outside of asymptotic regimes. In the simple case of a linear regression with missing values, the optimal predictor has a combinatorial expression: for $d$ features, there are $2^d$ possible missing-values patterns requiring $2^d$ models [13].

Le Morvan et al. [13] showed that in this setting, a multilayer perceptrons (MLP) can be consistent even in a pattern mixture MNAR model, but assuming $2^d$ hidden units. There have been many adaptations of neural networks to missing values, often involving an imputation with 0's and concatenating the mask (the indicator matrix coding for missing values) [23, 19, 15, 34, 4]. However there is no theory relating the network architecture to the impact of the missing-value mechanism on the prediction function. In particular, an important practical question is: how complex should the architecture be to cater for a given mechanism? Overly-complex architectures require a lot of data, but being too restrictive will introduce bias for missing values.

The present paper addresses the challenge of supervised learning with missing values. We propose a theoretically-grounded neural-network architecture which allows to implicitly impute values as a function of the observed data, aiming at the best prediction. More precisely,

- We derive an analytical expression of the Bayes predictor for linear regression in the presence of missing values under various missing data mechanisms including MAR and self-masking MNAR.
- We propose a new **principled** architecture, named NeuMiss network, based on a Neumann series approximation of the Bayes predictors, whose originality and strength is the use of $\odot M$ nonlinearities, i.e. the elementwise multiplication by the missingness indicator.
- We provide an upper bound on the Bayes risk of NeuMiss networks which highlights the benefits of depth and learning to approximate.
- We provide an interpretation of a classical ReLU network as a shallow NeuMiss network. We further demonstrate empirically the crucial role of the $\odot$ nonlinearities, by showing that increasing the capacity of NeuMiss networks improves predictions while it does not for classical networks.
- We show that NeuMiss networks are **suited medium-sized datasets**: they require $O(d^2)$ samples, contrary to $O(2^d)$ for methods that do not share weights between missing data patterns.
- We demonstrate the benefits of the proposed architecture over classical methods such as EM algorithms or iterative conditional imputation [31] both in terms of computational complexity –these methods scale in $O(2^d d^2)$ [28] and $O(d^3)$ respectively–, and in the ability to be **robust to the missing data mechanism**, including MNAR.

## 2 Optimal predictors in the presence of missing values

**Notations**  We consider a data set $\mathcal{D}_n = \{(X_1, Y_1), \ldots, (X_n, Y_n)\}$ of independent pairs $(X_i, Y_i)$, distributed as the generic pair $(X, Y)$, where $X \in \mathbb{R}^d$ and $Y \in \mathbb{R}$. We introduce the indicator vector $M \in \{0, 1\}^d$ which satisfies, for all $1 \le j \le d$, $M_j = 1$ if and only if $X_j$ is not observed. The random vector $M$ acts as a mask on $X$. We define the incomplete feature vector $\widetilde{X} \in \widetilde{\mathcal{X}} = (\mathbb{R} \cup \{\text{NA}\})^d$ (see [27], [26, appendix B]) as $\widetilde{X}_j = \text{NA}$ if $M_j = 1$, and $\widetilde{X}_j = X_j$ otherwise. As such, $\widetilde{X}$ is a

mixed categorical and continuous variable. An example of realization (lower-case letters) of the previous random variables would be a vector $x = (1.1, 2.3, -3.1, 8, 5.27)$ with the missing pattern $m = (0, 1, 0, 0, 1)$, giving $\widetilde{x} = (1.1, \text{ NA}, -3.1, 8, \text{ NA})$.

For realizations $m$ of $M$, we also denote by $obs(m)$ (resp. $mis(m)$) the indices of the zero entries of $m$ (resp. non-zero). Following classic missing-value notations, we let $X_{obs(M)}$ (resp. $X_{mis(M)}$) be the observed (resp. missing) entries in $X$. Pursuing the above example, we have $mis(m) = \{1, 4\}$, $obs(m) = \{0, 2, 3\}$, $x_{obs(m)} = (1.1, -3.1, 8)$, $x_{mis(m)} = (2.3, 5.27)$. To lighten notations, when there is no ambiguity, we remove the explicit dependence in $m$ and write, e.g., $X_{obs}$.

## 2.1 Problem statement: supervised learning with missing values

We consider a linear model of the complete data, such that the response $Y$ satisfies:

$$Y = \beta_0^\star + \langle X, \beta^\star \rangle + \varepsilon, \qquad \text{for some } \beta_0^\star \in \mathbb{R}, \beta^\star \in \mathbb{R}^d, \text{ and } \varepsilon \sim \mathcal{N}(0, \sigma^2). \tag{1}$$

Prediction with missing values departs from standard linear-model settings: the aim is to predict $Y$ given $\widetilde{X}$, as the complete input $X$ may be unavailable. The corresponding optimization problem is:

$$f_{\widetilde{X}}^\star \in \underset{f:\widetilde{\mathcal{X}} \to \mathbb{R}}{\operatorname{argmin}} \mathbb{E}[(Y - f(\widetilde{X}))^2], \tag{2}$$

where $f_{\widetilde{X}}^\star$ is the Bayes predictor for the squared loss, in the presence of missing values. The main difficulty of this problem comes from the half-discrete nature of the input space $\widetilde{\mathcal{X}}$. Indeed, the Bayes predictor $f_{\widetilde{X}}^\star(\widetilde{X}) = \mathbb{E}[Y \mid \widetilde{X}]$ can be rewritten as:

$$f_{\widetilde{X}}^\star(\widetilde{X}) = \mathbb{E}[Y \mid M, X_{obs(M)}] = \sum_{m \in \{0,1\}^d} \mathbb{E}[Y | X_{obs(m)}, M = m] \mathbb{1}_{M=m}, \tag{3}$$

which highlights the combinatorial issue of solving (2): one may need to optimize $2^d$ submodels, for the different $m$. In the following, we write the Bayes predictor $f^\star$ as a function of $(X_{obs(M)}, M)$:

$$f^\star(X_{obs(M)}, M) = \mathbb{E}[Y | X_{obs(M)}, M].$$

## 2.2 Expression of the Bayes predictor under various missing-values mechanisms

There is no general closed-form expression for the Bayes predictor, as it depends on the data distribution and missingness mechanism. However, an exact expression can be derived for Gaussian data with various missingness mechanisms.

**Assumption 1** (Gaussian data). *The distribution of $X$ is Gaussian, that is, $X \sim \mathcal{N}(\mu, \Sigma)$.*

**Assumption 2** (MCAR mechanism). *For all $m \in \{0,1\}^d$, $P(M = m|X) = P(M = m)$.*

**Assumption 3** (MAR mechanism). *For all $m \in \{0,1\}^d$, $P(M = m|X) = P(M = m|X_{obs(m)})$.*

**Proposition 2.1** (MAR Bayes predictor). *Assume that the data are generated via the linear model defined in equation (1) and satisfy Assumption 1. Additionally, assume that either Assumption 2 or Assumption 3 holds. Then the Bayes predictor $f^\star$ takes the form*

$$f^\star(X_{obs}, M) = \beta_0^\star + \langle \beta_{obs}^\star, X_{obs} \rangle + \langle \beta_{mis}^\star, \mu_{mis} + \Sigma_{mis,obs}(\Sigma_{obs})^{-1}(X_{obs} - \mu_{obs}) \rangle, \tag{4}$$

*where we use obs (resp. mis) instead of obs(M) (resp. mis(M)) for lighter notations.*

Obtaining the Bayes predictor expression turns out to be far more complicated for general MNAR settings but feasible for the Gaussian self-masking mechanism described below.

**Assumption 4** (Gaussian self-masking). *The missing data mechanism is self-masked with $P(M|X) = \prod_{k=1}^d P(M_k|X_k)$ and $\forall k \in [\![1, d]\!]$,*

$$P(M_k = 1|X_k) = K_k \exp\left(-\frac{1}{2}\frac{(X_k - \widetilde{\mu}_k)^2}{\widetilde{\sigma}_k^2}\right) \qquad \text{with } 0 < K_k < 1.$$

**Proposition 2.2** (Bayes predictor with Gaussian self-masking). *Assume that the data are generated via the linear model defined in equation* (1) *and satisfy Assumption 1 and Assumption 4. Let* $\Sigma_{mis|obs} = \Sigma_{mis,mis} - \Sigma_{mis,obs}\Sigma_{obs}^{-1}\Sigma_{obs,mis}$, *and let D be the diagonal matrix such that* $\mathrm{diag}(D) = (\widetilde{\sigma}_1^2, \ldots, \widetilde{\sigma}_d^2)$. *Then the Bayes predictor writes*

$$f^\star(X_{obs}, M) = \beta_0^\star + \langle \beta_{obs}^\star, X_{obs}\rangle + \langle \beta_{mis}^\star, (Id + D_{mis}\Sigma_{mis|obs}^{-1})^{-1}$$
$$\times (\tilde{\mu}_{mis} + D_{mis}\Sigma_{mis|obs}^{-1}(\mu_{mis} + \Sigma_{mis,obs}\left(\Sigma_{obs}\right)^{-1}(X_{obs} - \mu_{obs})))\rangle \quad (5)$$

The proof of Propositions 2.1 and 2.2 are in the Supplementary Materials (A.3 and A.4). These are the first results establishing exact expressions of the Bayes predictor in a MAR and specific MNAR mechanisms. Note that these propositions show that the Bayes predictor is linear by pattern under the assumptions studied, i.e., each of the $2^d$ submodels in equation 3 are linear functions of $X_{obs}$. For non-Gaussian data, the Bayes predictor may not be linear by pattern [13, Example 3.1].

**Generality of the Gaussian self-masking model**    For a self-masking mechanism where the probability of being missing increases (or decreases) with the value of the underlying variable, probit or logistic functions are often used [12]. A Gaussian self-masking model is also a suitable model: setting the mean of the Gaussian close to the extreme values gives a similar behaviour. In addition, it covers cases where the probability of being missing is centered around a given value.

# 3 NeuMiss networks: learning by approximating the Bayes predictors

## 3.1 Insight to build a network: sharing parameters across missing-value patterns

Computing the Bayes predictors in equations (4) or (5) requires to estimate the inverse of each submatrix $\Sigma_{obs(m)}$ for each missing-data pattern $m \in \{0,1\}^d$, *ie* one linear model per missing-data pattern. For a number of hidden units $\propto 2^d$, a MLP with ReLU non-linearities can fit these linear models independently from one-another, and is shown to be consistent [13]. But it is prohibitive when $d$ grows. Such an architecture is largely over-parametrized as it does not share information between similar missing-data patterns. Indeed, the slopes of each of the linear regression per pattern given by the Bayes predictor in equations (4) and (5) are linked via the inverses of $\Sigma_{obs}$.

Thus, one approach is to estimate only one vector $\mu$ and one covariance matrix $\Sigma$ via an expectation maximization (EM) algorithm [2], and then compute the inverses of $\Sigma_{obs}$. But the computational complexity then scales linearly in the number of missing-data patterns (which is in the worst case exponential in the dimension $d$), and is therefore also prohibitive when the dimension increases.

In what follows, we propose an in-between solution, modeling the relationships between the slopes for different missing-data patterns without directly estimating the covariance matrix. Intuitively, observations from one pattern will be used to estimate the regression parameters of other patterns.

## 3.2 Differentiable approximations of the inverse covariances with Neumann series

The major challenge of equations (4) and (5) is the inversion of the matrices $\Sigma_{obs(m)}$ for all $m \in \{0,1\}^d$. Indeed, there is no simple relationship for the inverses of different submatrices in general. As a result, the slope corresponding to a pattern $m$ cannot be easily expressed as a function of $\Sigma$.

We therefore propose to approximate $\left(\Sigma_{obs(m)}\right)^{-1}$ for all $m \in \{0,1\}^d$ recursively in the following way. First, we choose as a starting point a $d \times d$ matrix $S^{(0)}$. $S_{obs(m)}^{(0)}$ is then defined as the sub-matrix of $S^{(0)}$ obtained by selecting the columns and rows that are observed (components for which $m = 0$) and is our order-0 approximation of $\left(\Sigma_{obs(m)}\right)^{-1}$. Then, for all $m \in \{0,1\}^d$, we define the order-$\ell$ approximation $S_{obs(m)}^{(\ell)}$ of $\left(\Sigma_{obs(m)}\right)^{-1}$ via the following iterative formula: for all $\ell \geq 1$,

$$S_{obs(m)}^{(\ell)} = (Id - \Sigma_{obs(m)})S_{obs(m)}^{(\ell-1)} + Id. \quad (6)$$

The iterates $S_{obs(m)}^{(\ell)}$ converge linearly to $(\Sigma_{obs(m)})^{-1}$ (A.5 in the Supplementary Materials), and are in fact Neumann series truncated to $\ell$ terms if $S^{(0)} = Id$.

We now define the order-$\ell$ approximation of the Bayes predictor in MAR settings (equation (4)) as

$$f_\ell^\star(X_{obs}, M) = \langle \beta_{obs}^\star, X_{obs} \rangle + \langle \beta_{mis}^\star, \mu_{mis} + \Sigma_{mis,obs} S_{obs(m)}^{(\ell)} (X_{obs} - \mu_{obs}) \rangle. \qquad (7)$$

The error between the Bayes predictor and its order-$\ell$ approximation is provided in Proposition 3.1.

**Proposition 3.1.** *Let $\nu$ be the smallest eigenvalue of $\Sigma$. Assume that the data are generated via a linear model defined in equation (1) and satisfy Assumption 1. Additionally, assume that either Assumption 2 or Assumption 3 holds and that the spectral radius of $\Sigma$ is strictly smaller than one. Then, for all $\ell \geq 1$,*

$$\mathbb{E}\left[ \left( f_\ell^\star(X_{obs}, M) - f^\star(X_{obs}, M) \right)^2 \right] \leq \frac{(1-\nu)^{2\ell} \|\beta^\star\|_2^2}{\nu} \mathbb{E}\left[ \left\| Id - S_{obs(M)}^{(0)} \Sigma_{obs(M)} \right\|_2^2 \right] \qquad (8)$$

The error of the order-$\ell$ approximation decays exponentially fast with $\ell$. More importantly, if the submatrices $S_{obs}^{(0)}$ of $S^{(0)}$ are good approximations of $(\Sigma_{obs})^{-1}$ on average, that is if we choose $S^{(0)}$ which minimizes the expectation in the right-hand side in inequality (8), then our model provides a good approximation of the Bayes predictor even with order $\ell = 0$. This is the case for a diagonal covariance matrix, as taking $S^{(0)} = \Sigma^{-1}$ has no approximation error as $(\Sigma^{-1})_{obs} = (\Sigma_{obs})^{-1}$.

### 3.3 NeuMiss network architecture: multiplying by the mask

**Network architecture**   We propose a neural-network architecture to approximate the Bayes predictor, where the inverses $(\Sigma_{obs})^{-1}$ are computed using an unrolled version of the iterative algorithm. Figure 1 gives a diagram for such neural network using an order-3 approximation corresponding to a depth 4. $x$ is the input, with missing values replaced by 0. $\mu$ is a trainable parameter corresponding to the parameter $\mu$ in equation (7). To match the Bayes predictor exactly (equation (7)), weight matrices should be simple transformations of the covariance matrix indicated in blue on Figure 1.

Following strictly Neummann iterates would call for a shared weight matrix across all $W_{Neu}^{(k)}$. Rather, we learn each layer independently. This choice is motivated by works on iterative algorithm unrolling [5] where independent layers' weights can improve a network's approximation performance [33]. Note that [3] has also introduced a neural network architecture based on unrolling the Neumann series. However, their goal is to solve a linear inverse problem with a learned regularization, which is very different from ours.

**Multiplying by the mask**   Note that the observed indices change for each sample, leading to an implementation challenge. For a sample with missing data pattern $m$, the weight matrices $S^{(0)}, W_{Neu}^{(1)}$ and $W_{Neu}^{(2)}$ of Figure 1 should be masked such that their rows and columns corresponding to the indices $mis(m)$ are zeroed, and the rows of $W_{Mix}$ corresponding to $obs(m)$ as well as the columns of $W_{Mix}$ corresponding to $mis(m)$ are zeroed. Implementing efficiently a network in which the weight matrices are masked differently for each sample can be challenging. We thus use the following trick. Let $W$ be a weight matrix, $v$ a vector, and $\bar{m} = 1 - m$. Then $(W \odot \bar{m}\bar{m}^\top)v = (W(v \odot \bar{m})) \odot \bar{m}$, i.e, using a masked weight matrix is equivalent to masking the input and output vector. The network can then be seen as a classical network where the nonlinearities are multiplications by the mask.

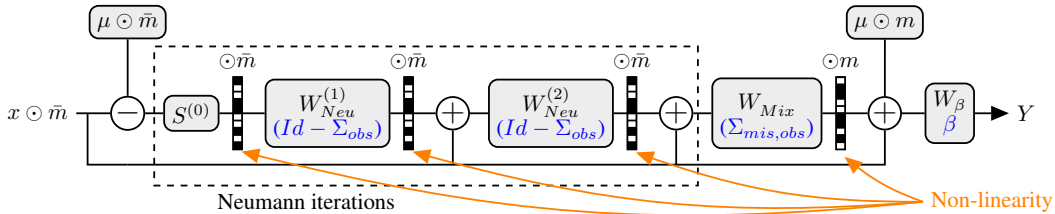

Figure 1: **NeuMiss network architecture with a depth of 4** — $\bar{m} = 1 - m$. Each weight matrix $W^{(k)}$ corresponds to a simple transformation of the covariance matrix indicated in blue.

**Approximation of the Gaussian self-masking Bayes predictor**  Although our architecture is motivated by the expression of the Bayes predictor in MCAR and MAR settings, a similar architecture can be used to target the prediction function (5) for self-masking data. To see why, let's first assume that $D_{mis}\Sigma_{mis|obs}^{-1} \approx Id$. Then, the self-masking Bayes predictor (5) becomes:

$$f^\star(X_{obs}, M) \approx \beta_0^\star + \langle \beta_{obs}^\star, X_{obs} \rangle$$
$$+ \langle \beta_{mis}^\star, \frac{1}{2}(\tilde{\mu}_{mis} + \mu_{mis}) + \frac{1}{2}\Sigma_{mis,obs}(\Sigma_{obs})^{-1}(X_{obs} - \mu_{obs}) \rangle \quad (9)$$

i.e., its expression is the same as for the M(C)AR Bayes predictor (4) except that $\mu_{mis}$ is replaced by $\frac{1}{2}(\tilde{\mu}_{mis} + \mu_{mis})$ and $\Sigma_{mis,obs}$ is scaled down by a factor $\frac{1}{2}$. Thus, under this approximation, the self-masking Bayes predictor can be modeled by our proposed architecture (just as the M(C)AR Bayes predictor), the only difference being the targeted values for the parameters $\mu$ and $W_{mix}$ of the network. A less coarse approximation also works: $D_{mis}\Sigma_{mis|obs}^{-1} \approx \hat{D}_{mis}$ where $\hat{D}$ is a diagonal matrix. In this case, the proposed architecture can perfectly model the self-masking Bayes predictor: the parameter $\mu$ of the network should target $(Id + \hat{D})^{-1}(\tilde{\mu} + \hat{D}\mu)$ and $W_{mix}$ should target $(Id + \hat{D})^{-1}\hat{D}\Sigma$ instead of simply $\Sigma$ in the M(C)AR case. Consequently, our architecture can well approximate the self-masking Bayes predictor by adjusting the values learned for the parameters $\mu$ and $W_{mix}$ if $D_{mis}\Sigma_{mis|obs}^{-1}$ are close to diagonal matrices.

### 3.4  Link with the multilayer perceptron with ReLU activations

A common practice to handle missing values is to consider as input the data concatenated with the mask *eg* in [13]. The next proposition connects this practice to Neumman networks.

**Proposition 3.2** (equivalence MLP - depth-1 NeuMiss network). *Let $[X \odot (1 - M), M] \in [0, 1]^d \times \{0, 1\}^d$ be an input $X$ imputed by 0 concatenated with the mask $M$.*

- *Let $\mathcal{H}_{ReLU} = (W \in \mathbb{R}^{d \times 2d}, ReLU)$ be a hidden layer which connects $[X \odot (1 - M), M]$ to $d$ hidden units, and applies a ReLU nonlinearity to the activations.*
- *Let $\mathcal{H}_{\odot M} = (W \in \mathbb{R}^{d \times d}, \mu, \odot M)$ be a hidden layer that connects an input $(X - \mu) \odot (1 - M)$ to $d$ hidden units, and applies a $\odot M$ nonlinearity.*

*Denote by $h_k^{ReLU}$ and $h_k^{\odot M}$ the outputs of the $k^{th}$ hidden unit of each layer. Then there exists a configuration of the weights of the hidden layer $\mathcal{H}_{ReLU}$ such that $\mathcal{H}_{\odot M}$ and $\mathcal{H}_{ReLU}$ have the same hidden units activated for any $(X_{obs}, M)$, and activated hidden units are such that $h_k^{ReLU}(X_{obs}, M) = h_k^{\odot M}(X_{obs}, M) + c_k$ where $c_k \in \mathbb{R}$.*

Proposition 3.2 states that a hidden layer $\mathcal{H}_{ReLU}$ can be rewritten as a $\mathcal{H}_{\odot M}$ layer up to a constant. Note that, as soon as another layer is stacked after $\mathcal{H}_{\odot M}$ or $\mathcal{H}_{ReLU}$, this additional constant can be absorbed into the biases of this new layer. Thus the weights of $\mathcal{H}_{ReLU}$ can be learned so as to mimic $\mathcal{H}_{\odot M}$. In our case, this means that a MLP with ReLU activations, one hidden layer of $d$ hidden units, and which operates on the concatenated vector, is closely related to the 1-depth NeuMiss network (see Figure 1), thereby providing theoretical support for the use of the latter MLP. This theoretical link completes the results of [13], who showed experimentally that in such a MLP $O(d)$ units were enough to perform well on Gaussian data, but only provided theoretical results with $2^d$ hidden units.

## 4  Empirical results

### 4.1  The $\odot M$ nonlinearity is crucial to the performance

The specificity of NeuMiss networks resides in the $\odot M$ nonlinearities, instead of more conventional choices such as ReLU. Figure 2 shows how the choice of nonlinearity impacts the performance as a function of the depth. We compare two networks that take as input the data imputed by 0 concatenated with the mask: MLP Deep which has 1 to 10 hidden layers of $d$ hidden units followed by ReLU nonlinearities and MLP Wide which has one hidden layer whose width is increased followed by a ReLU nonlinearity. This latter was shown to be consistent given $2^d$ hidden units [13].

Figure 2 shows that increasing the capacity (depth) of MLP Deep fails to improve the performances, unlike with NeuMiss networks. Similarly, it is also significantly more effective to increase the

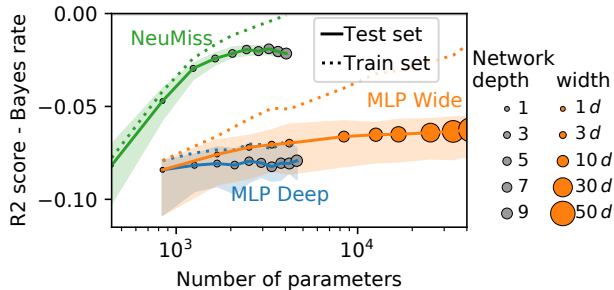

Figure 2: **Performance as a function of capacity across architectures** — Empirical evolution of the performance for a linear generating mechanism in MCAR settings. Data are generated under a linear model with Gaussian covariates in a MCAR setting (50% missing values, $n = 10^5$, $d = 20$).

capacity of the NeuMiss network (depth) than to increase the capacity (width) of MLP Wide. These results highlight the crucial role played by the $\odot$ nonlinearity. Finally, the performance of MLP Wide with $d$ hidden units is close to that of NeuMiss with a depth of 1, suggesting that it may rely on the weight configuration established in Proposition 3.2.

## 4.2 Approximation learned by the NeuMiss network

The NeuMiss architecture was designed to approximate well the Bayes predictor (4). As shown in Figure 1, its weights can be chosen so as to express the Neumann approximation of the Bayes predictor (7) exactly. We will call this particular instance of the network, with $S^{(0)}$ set to identity, the analytic network. However, just like LISTA [5] learns improved weights compared to the ISTA iterations, the NeuMiss network may learn improved weights compared to the Neumann iterations. Comparing the performance of the analytic network to its learned counterpart on simulated MCAR data, Figure 3 (left) shows that the learned network requires a much smaller depth compared to the analytic network to reach a given performance. Moreover, the depth-1 learned network largely outperforms the depth-1 analytic network, which means that it is able to learn a good initialization $S^{(0)}$ for the iterates. Figure 3 also compares the performance of the learned network with and without residual connections, and shows that residual connections are not needed for good performance. This observation is another hint that the iterates learned by the network depart from the Neumann ones.

## 4.3 NeuMiss networks require $O(d^2)$ samples

Figure 3 (right) studies the depth for which NeuMiss networks perform well for different number of samples $n$ and features $d$. It outlines that NeuMiss networks work well in regimes with more than 10 samples available per model parameters, where the number of model parameters scales as $d^2$. In general, even with many samples, depth of more than 5 explore diminishing returns. Supplementary figure 5 shows the same behavior in various MNAR settings.

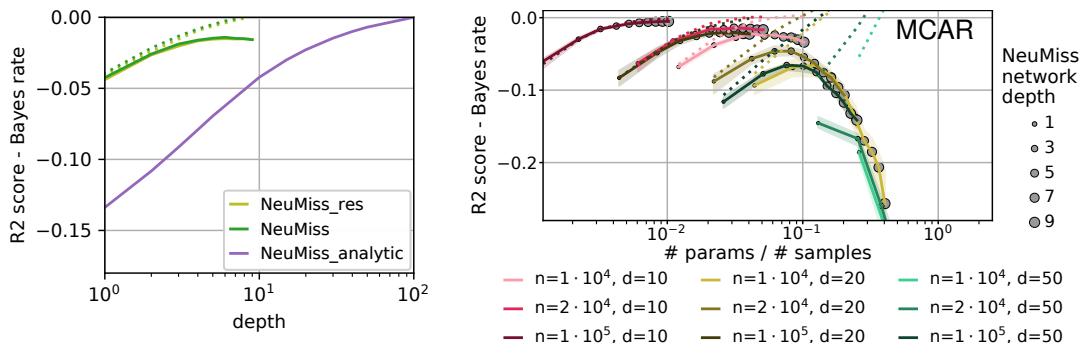

Figure 3: **Left: learned versus analytic Neumann iterates** — NeuMiss analytic is the NeuMiss architecture with weights set to represent (6), supposing we have access to the ground truth parameters, NeuMiss (resp. NeuMiss res) corresponds to the network without (resp. with) residual connections. **Right: Required capacity in various settings** — Performance of NeuMiss networks varying the depth in simulations with different number of samples $n$ and of features $d$.

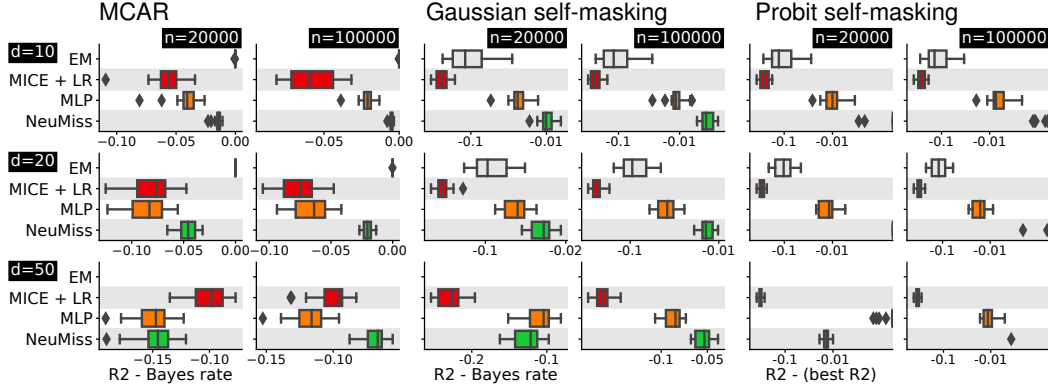

Figure 4: **Predictive performances in various scenarios** — varying missing-value mechanisms, number of samples $n$, and number of features $d$. All experiments are repeated 20 times. For self-masking settings, the x-xaxis is in log scale, to accommodate the large difference between methods.

### 4.4 Prediction performance: NeuMiss networks are robust to the missing data mechanism

We now evaluate the performance of NeuMiss networks compared to other methods under various missing values mechanisms. The data are generated according to a multivariate Gaussian distribution, with a covariance matrix $\Sigma = UU^\top + \text{diag}(\epsilon)$, $U \in \mathbb{R}^{d \times \frac{d}{2}}$, and the entries of $U$ drawn from a standard normal distribution. The noise $\epsilon$ is a vector of entries drawn uniformly in $\left[10^{-2}, 10^{-1}\right]$ to make $\Sigma$ full rank. The mean is drawn from a standard normal distribution. The response $Y$ is generated as a linear function of the complete data $X$ as in equation 1. The noise is chosen to obtain a signal-to-noise ratio of 10. 50% of entries on each features are missing, with various missing data mechanisms: MCAR, MAR, Gaussian self-masking and Probit self-masking. The Gaussian self-masking is obtained according to Assumption 4, while the Probit self-masking is a similar setting where the probability for feature $j$ to be missing depends on its value $X_j$ through an inverse probit function. We compare the performances of the following methods:

- **EM**: an Expectation-Maximisation algorithm [30] is run to estimate the parameters of the joint probability distribution of $X$ and $Y$ –Gaussian– with missing values. Then based on this estimated distribution, the prediction is given by taking the expectation of $Y$ given $X$.

- **MICE + LR**: the data is first imputed using conditional imputation as implemented in scikit-learn's [25] IterativeImputer, which proceeds by iterative ridge regression. It adapts the well known MICE [31] algorithm to be able to impute a test set. A linear regression is then fit on the imputed data.

- **MLP**: A multilayer perceptron as in [13], with one hidden layer followed by a ReLU nonlinearity, taking as input the data imputed by 0 concatenated with the mask. The width of the hidden layer is varied between $d$ and $100\,d$ hidden units, and chosen using a validation set. The MLP is trained using ADAM and a batch size of 200. The learning rate is initialized to $\frac{10^{-2}}{d}$ and decreased by a factor of 0.2 when the loss stops decreasing for 2 epochs. The training finishes when either the learning rate goes below $5 \times 10^{-6}$ or the maximum number of epochs is reached.

- **NeuMiss** : The NeuMiss architecture, without residual connections, choosing the depth on a validation set. The architecture was implemented using PyTorch [24], and optimized using stochastic gradient descent and a batch size of 10. The learning rate schedule and stopping criterion are the same as for the MLP.

For MCAR, MAR, and Gaussian self-masking settings, the performance is given as the obtained R2 score minus the Bayes rate (the closer to 0 the better), the best achievable R2 knowing the underlying ground truth parameters. In our experiments, an estimation of the Bayes rate is obtained using the score of the Bayes predictor. For probit self-masking, as we lack an analytical expression for the Bayes predictor, the performance is given with respect to the best performance achieved across all methods. The code to reproduce the experiments is available in GitHub [1].

In MCAR settings, figure 4 shows that, as expected, EM gives the best results when tractable. Yet, we could not run it for number of features $d \geq 50$. NeuMiss is the best performing method behind EM, in all cases except for $n = 2 \times 10^4, d = 50$, where depth of 1 or greater overfit due to the low ratio of number of parameters to number of samples. In such situation, MLP has the same expressive power and performs slightly better. Note that for a high samples-to-parameters ratio ($n = 1 \times 10^5, d = 10$), NeuMiss reaches an almost perfect $R2$ score, less than 1% below the Bayes rate. The results for the MAR setting are very similar to the MCAR results, and are given in supplementary figure 6.

For the self-masking mechanisms, the NeuMiss network significantly improves upon the competitors, followed by the MLP. This is even true for the probit self-masking case for which we have no theoretical results. The gap between the two architectures widens as the number of samples increases, with the NeuMiss network benefiting from a large amount of data. These results emphasize the robustness of NeuMiss and MLP to the missing data mechanism, including MNAR settings in which EM or conditional imputation do not enable statistical analysis.

## 5 Discussion and conclusion

Traditionally, statistical models are adapted to missing values using EM or imputation. However, these require strong assumptions on the missing values. Rather, we frame the problem as a risk minimization with a flexible yet tractable function family. We propose the NeuMiss network, a theoretically-grounded architecture that handles missing values using multiplication by the mask as nonlinearities. It targets the Bayes predictor with differentiable approximations of the inverses of the various covariance submatrices, thereby reducing complexity by sharing parameters across missing data patterns. Strong connections between a shallow version of our architecture and the common practice of inputing the mask to an MLP is established.

The NeuMiss architecture has clear practical benefits. It is robust to the missing-values mechanism, often unknown in practice. Moreover its sample and computational complexity are independent of the number of missing-data patterns, which allows to work with datasets of higher dimensionality and limited sample sizes. This work opens many perspectives, in particular using this network as a building block in larger architectures, *eg* to tackle nonlinear problems.

## Broader Impact

In our work, we proposed theoretical foundations to justify the use of a specific neural network architecture in the presence of missing-values.

Neural networks are known for their challenging black-box nature. We believe that such theory leads to a better understanding of the mechanisms at work in neural networks.

Our architecture is tailored for missing data. These are present in many applications, in particular in social or health data. In these fields, it is common for under-represented groups to exhibit a higher percentage of missing values (MNAR mechanism). Dealing with these missing values will definitely improve prediction for these groups, thereby reducing potential bias against these exact same groups.

As any predictive algorithm, our proposal can be misused in a variety of context, including in medical science, for which a proper assessment of the specific characteristics of the algorithm output is required (assessing bias in prediction, prevent false conclusion resulting from misinterpreting outputs). Yet, by improving performance and understanding of a fundamental challenge in many applications settings, our work is not facilitating more unethical aspects of AI than ethical applications. Rather, medical studies that suffer chronically from limited sample sizes are mostly likely to benefit from the reduced sample complexity that these advances provide.

## Acknowledgments and Disclosure of Funding

This work was funded by ANR-17-CE23-0018 - DirtyData - Intégration et nettoyage de données pour l'analyse statistique (2017) and the MissingBigData grant from DataIA.

## Footnotes

[1]https://github.com/marineLM/NeuMiss

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
