[Supplementary Material]

# Supplementary materials – NeuMiss networks: differentiable programming for supervised learning with missing values

## A Proofs

### A.1 Proof of Lemma 1

**Lemma 1** (General expression of the Bayes predictor). *Assume that the data are generated via the linear model defined in equation (1), then the Bayes predictor takes the form*

$$f^\star(X_{obs(M)}, M) = \beta_0^\star + \langle \beta_{obs(M)}^\star, X_{obs(M)} \rangle + \langle \beta_{mis(M)}^\star, \mathbb{E}[X_{mis(M)} | M, X_{obs(M)}] \rangle, \tag{10}$$

*where $(\beta_{obs(M)}^\star, \beta_{mis(M)}^\star)$ correspond to the decomposition of the regression coefficients in observed and missing elements.*

*Proof of Lemma 1.* By definition of the linear model, we have

$$\begin{aligned}
f_{\widetilde{X}}^\star(\widetilde{X}) &= \mathbb{E}[Y|\widetilde{X}] \\
&= \mathbb{E}[\beta_0^\star + \langle \beta^\star, X \rangle \mid M, X_{obs(M)}] \\
&= \beta_0^\star + \langle \beta_{obs(M)}^\star, X_{obs(M)} \rangle + \langle \beta_{mis(M)}^\star, \mathbb{E}[X_{mis(M)} \mid M, X_{obs(M)}] \rangle.
\end{aligned}$$

$\square$

### A.2 Proof of Lemma 2

**Lemma 2** (Product of two multivariate gaussians). *Let $f(X) = \exp\left((X-a)^\top A^{-1}(X-a)\right)$ and $g(X) = \exp\left((X-b)^\top B^{-1}(X-b)\right)$ be two Gaussian functions, with $A$ and $B$ positive semidefinite matrices. Then the product $f(X)g(X)$ is another gaussian function given by:*

$$f(X)g(X) = \exp\left(-\frac{1}{2}(a-b)^\top (A+B)^{-1}(a-b))\right) \exp\left(-\frac{1}{2}(X-\mu_p)^\top \Sigma_p^{-1}(X-\mu_p)\right)$$

*where $\mu_p$ and $\Sigma_p$ depend on a, A, b and B.*

*Proof of Lemma 2.* Identifying the second and first order terms in $X$ we get:

$$\Sigma_p^{-1} = A^{-1} + B^{-1} \tag{11}$$

$$\Sigma_p^{-1}\mu_p = A^{-1}a + B^{-1}b \tag{12}$$

By completing the square, the product can be rewritten as:

$$f(X)g(X) = \exp\left(-\frac{1}{2}(a^\top A^{-1}a + b^\top B^{-1}b - \mu_p^\top \Sigma_p^{-1}\mu_p)\right) \exp\left(-\frac{1}{2}(X-\mu_p)^\top \Sigma_p^{-1}(X-\mu_p)\right)$$

Let's now simplify the scaling factor:

$$\begin{aligned}
c &= a^\top A^{-1}a + b^\top B^{-1}b - \mu_p^\top \Sigma_p^{-1}\mu_p \\
&= a^\top A^{-1}a + b^\top B^{-1}b - \left(a^\top A^{-1}(A^{-1}+B^{-1})^{-1} + b^\top B^{-1}(A^{-1}+B^{-1})^{-1}\right)\left(A^{-1}a + B^{-1}b\right) \\
&= a^\top (A^{-1} - A^{-1}(A^{-1}+B^{-1})^{-1}A^{-1})a + b^\top (B^{-1} - B^{-1}(A^{-1}+B^{-1})^{-1}B^{-1})b \\
&\quad - 2a^\top (A^{-1}(A^{-1}+B^{-1})^{-1}B^{-1})b \\
&= a^\top (A+B)^{-1}a + b^\top (A+B)^{-1}b - 2a^\top (A+B)^{-1}b \\
&= (a-b)^\top (A+B)^{-1}(a-b)
\end{aligned}$$

The third equality is true because $A$ and $B$ are symmetric. The fourth equality uses the Woodbury identity and the fact that:

$$\begin{aligned}
(A^{-1}(A^{-1}+B^{-1})^{-1}B^{-1}) &= \left(B(A^{-1}+B^{-1})A\right)^{-1} \\
&= \left(BA^{-1}A + BB^{-1}A\right)^{-1} \\
&= (B+A)^{-1}
\end{aligned}$$

The last equality allows to conclude the proof. $\square$

## A.3 Proof of Proposition 2.1

**Proposition 2.1** (MAR Bayes predictor). *Assume that the data are generated via the linear model defined in equation (1) and satisfy Assumption 1. Additionally, assume that either Assumption 2 or Assumption 3 holds. Then the Bayes predictor $f^\star$ takes the form*

$$f^\star(X_{obs}, M) = \beta_0^\star + \langle \beta_{obs}^\star, X_{obs} \rangle + \langle \beta_{mis}^\star, \mu_{mis} + \Sigma_{mis,obs}(\Sigma_{obs})^{-1}(X_{obs} - \mu_{obs}) \rangle, \quad (4)$$

*where we use obs (resp. mis) instead of obs(M) (resp. mis(M)) for lighter notations.*

Lemma 1 gives the general expression of the Bayes predictor for any data distribution and missing data mechanism. From this expression, on can see that the crucial step to compute the Bayes predictor is computing $\mathbb{E}[X_{mis}|M, X_{obs}]$, or in other words, $\mathbb{E}[X_j|M, X_{obs}]$ for all $j \in mis$. In order to compute this expectation, we will characterize the distribution $P(X_j|M, X_{obs})$ for all $j \in mis$. Let $mis'(M, j) = mis(M) \setminus \{j\}$. For clarity, when there is no ambiguity we will just write $mis'$. Using the sum and product rules of probability, we have:

$$P(X_j|M, X_{obs}) = \frac{P(M, X_j, X_{obs})}{P(M, X_{obs})} \quad (13)$$

$$= \frac{\int P(M, X_j, X_{obs}, X_{mis'}) \mathrm{d}X_{mis'}}{\int \int P(M, X_j, X_{obs}, X_{mis'}) \mathrm{d}X_{mis'} \mathrm{d}X_j} \quad (14)$$

$$= \frac{\int P(M|X_{obs}, X_j, X_{mis'}) P(X_{obs}, X_j, X_{mis'}) \mathrm{d}X_{mis'}}{\int \int P(M|X_{obs}, X_j, X_{mis'}) P(X_{obs}, X_j, X_{mis'}) \mathrm{d}X_{mis'} \mathrm{d}X_j} \quad (15)$$

In the MCAR case, for all $m \in \{0, 1\}^d$, $\mathbb{P}(M = m|X) = \mathbb{P}(M = m)$, thus we have

$$P(X_j|M, X_{obs}) = \frac{P(M) \int P(X_{obs}, X_j, X_{mis'}) \mathrm{d}X_{mis'}}{P(M) \int \int P(X_{obs}, X_j, X_{mis'}) \mathrm{d}X_{mis'} \mathrm{d}X_j} \quad (16)$$

$$= \frac{P(X_{obs}, X_j)}{P(X_{obs})} \quad (17)$$

$$= P(X_j|X_{obs}) \quad (18)$$

On the other hand, assuming MAR mechanism, that is, for all $m \in \{0, 1\}^d$, $P(M = m|X) = P(M = m|X_{obs(m)})$, we have, given equation (15),

$$P(X_j|M, X_{obs}) = \frac{P(M|X_{obs}) \int P(X_{obs}, X_j, X_{mis'}) \mathrm{d}X_{mis'}}{P(M|X_{obs}) \int \int P(X_{obs}, X_j, X_{mis'}) \mathrm{d}X_{mis'} \mathrm{d}X_j} \quad (19)$$

$$= \frac{P(X_{obs}, X_j)}{P(X_{obs})} \quad (20)$$

$$= P(X_j|X_{obs}) \quad (21)$$

Therefore, if the missing data mechanism is MCAR or MAR, we have, according to equation (18) and (21),

$$\mathbb{E}[X_{mis(M)} \mid M, X_{obs(M)}] = \mathbb{E}[X_{mis(M)} \mid X_{obs(M)}].$$

Since $X$ is a Gaussian vector distributed as $\mathcal{N}(\mu, \Sigma)$, we know that the conditional expectation $\mathbb{E}[X_{mis(M)} \mid X_{obs(M)}]$ satisfies

$$\mathbb{E}\left[X_{mis(m)} \mid X_{obs(m)}\right] = \mu_{mis(m)} + \Sigma_{mis(m),obs(m)} \left(\Sigma_{obs(m)}\right)^{-1} \left(X_{obs(m)} - \mu_{obs(m)}\right), \quad (22)$$

[see, e.g., 17]. This concludes the proof according to Lemma 1.

## A.4 Proof of Proposition 2.2

**Proposition 2.2** (Bayes predictor with Gaussian self-masking). *Assume that the data are generated via the linear model defined in equation (1) and satisfy Assumption 1 and Assumption 4. Let $\Sigma_{mis|obs} = \Sigma_{mis,mis} - \Sigma_{mis,obs}\Sigma_{obs}^{-1}\Sigma_{obs,mis}$, and let $D$ be the diagonal matrix such that $\mathrm{diag}(D) = (\tilde{\sigma}_1^2, \ldots, \tilde{\sigma}_d^2)$. Then the Bayes predictor writes*

$$f^\star(X_{obs}, M) = \beta_0^\star + \langle \beta_{obs}^\star, X_{obs} \rangle + \langle \beta_{mis}^\star, (Id + D_{mis}\Sigma_{mis|obs}^{-1})^{-1}$$
$$\times (\tilde{\mu}_{mis} + D_{mis}\Sigma_{mis|obs}^{-1}(\mu_{mis} + \Sigma_{mis,obs}(\Sigma_{obs})^{-1}(X_{obs} - \mu_{obs}))) \rangle \quad (5)$$

In the Gaussian self-masking case, according to Assumption 4, the probability factorizes as $P(M = m|X) = P(M_{mis(m)} = 1|X_{mis(m)})P(M_{obs(m)} = 0|X_{obs(m)})$. Equation 15 can thus be rewritten as:

$$P(X_j|M, X_{obs}) = \frac{P(M_{obs} = 0|X_{obs}) \int P(M_{mis} = 1|X_{mis})P(X_{obs}, X_j, X_{mis'}) \mathrm{d}X_{mis'}}{P(M_{obs} = 0|X_{obs}) \int \int P(M_{mis} = 1|X_{mis})P(X_{obs}, X_j, X_{mis'}) \mathrm{d}X_{mis'} \mathrm{d}X_j} \quad (23)$$

$$= \frac{\int P(M_{mis} = 1|X_{mis})P(X_{mis}|X_{obs}) \mathrm{d}X_{mis'}}{\int \int P(M_{mis} = 1|X_{mis})P(X_{mis}|X_{obs}) \mathrm{d}X_{mis'} \mathrm{d}X_j} \quad (24)$$

Let $D$ be the diagonal matrix such that $\mathrm{diag}(D) = \widetilde{\sigma}^2$, where $\widetilde{\sigma}$ is defined in Assumption 4. Then the masking probability reads:

$$P(M_{mis} = 1|X_{mis}) = \prod_{k \in mis}^{d} K_k \exp\left(-\frac{1}{2}(X_{mis} - \widetilde{\mu}_{mis})(D_{mis,mis})^{-1}(X_{mis} - \widetilde{\mu}_{mis})\right) \tag{25}$$

Using the conditional Gaussian formula, we have $P(X_{mis}|X_{obs}) = \mathcal{N}(X_{mis}|\mu_{mis|obs}, \Sigma_{mis|obs})$ with

$$\mu_{mis|obs} = \mu_{mis} + \Sigma_{mis,obs}\Sigma_{obs,obs}^{-1}(X_{obs} - \mu_{obs}) \tag{26}$$

$$\Sigma_{mis|obs} = \Sigma_{mis,mis} - \Sigma_{mis,obs}\Sigma_{obs}^{-1}\Sigma_{obs,mis} \tag{27}$$

Thus, according to equation (25), $P(M_{mis} = 1|X_{mis})$ and $P(X_{mis}|X_{obs})$ are Gaussian functions of $X_{mis}$. By Lemma 2, their product is also a Gaussian function given by:

$$P(M_{mis} = 1|X_{mis})P(X_{mis}|X_{obs}) = K \exp\left(-\frac{1}{2}(X_{mis} - a_M)^\top (A_M)^{-1}(X_{mis} - a_M)\right) \tag{28}$$

where $a_M$ and $A_M$ depend on the missingness pattern and

$$K = \prod_{k \in mis}^{d} \frac{K_k}{\sqrt{(2\pi)^{|mis|}|\Sigma_{mis|obs}|}} \exp\left(-\frac{1}{2}(\widetilde{\mu}_{mis} - \mu_{mis|obs})^\top (\Sigma_{mis|obs} + D_{mis,mis})^{-1}(\widetilde{\mu}_{mis} - \mu_{mis|obs})\right) \tag{29}$$

$$(A_M)^{-1} = D_{mis,mis}^{-1} + \Sigma_{mis|obs}^{-1} \tag{30}$$

$$(A_M)^{-1} a_M = D_{mis,mis}^{-1}\widetilde{\mu}_{mis} + \Sigma_{mis|obs}^{-1}\mu_{mis|obs} \tag{31}$$

Because $K$ does not depend on $X_{mis}$, it simplifies from eq 24. As a result we get:

$$P(X_j|M, X_{obs}) = \frac{\int \mathcal{N}(X_{mis}|a_M, A_M)\mathrm{d}X_{mis'}}{\int\int \mathcal{N}(X_{mis}|a_M, A_M)\mathrm{d}X_{mis'}\mathrm{d}X_j} \tag{32}$$

$$= \mathcal{N}(X_j|(a_M)_j, (A_M)_{j,j}) \tag{33}$$

By definition of the Bayes predictor, we have

$$f_{\widetilde{X}}^\star(\widetilde{X}) = \beta_0^\star + \langle\beta_{obs(M)}^\star, X_{obs(M)}\rangle + \langle\beta_{mis(M)}^\star, \mathbb{E}[X_{mis(M)}|M, X_{obs(M)}]\rangle, \tag{34}$$

where

$$\mathbb{E}[X_{mis}|M, X_{obs}] = (a_M)_{mis}. \tag{35}$$

Combining equations (30), (31), (35), we obtain

$$\mathbb{E}[X_{mis}|M, X_{obs}] = \left(Id + D_{mis}\Sigma_{mis|obs}^{-1}\right)^{-1} \tag{36}$$

$$\times \left[\widetilde{\mu}_{mis} + D_{mis}\Sigma_{mis|obs}^{-1}\left(\mu_{mis} + \Sigma_{mis,obs}(\Sigma_{obs})^{-1}(X_{obs} - \mu_{obs})\right)\right] \tag{37}$$

## A.5 Controlling the convergence of Neumann iterates

Here we establish an auxiliary result, controlling the convergence of Neumann iterates to the matrix inverse.

**Proposition A.1** (Linear convergence of Neumann iterations). *Assume that the spectral radius of $\Sigma$ is strictly less than 1. Therefore, for all missing data patterns $m \in \{0,1\}^d$, the iterates $S_{obs(m)}^{(\ell)}$ defined in equation (6) converge linearly towards $(\Sigma_{obs(m)})^{-1}$ and satisfy, for all $\ell \geq 1$,*

$$\|Id - \Sigma_{obs(m)}S_{obs(m)}^{(\ell)}\|_2 \leq (1 - \nu_{obs(m)})^\ell\|Id - \Sigma_{obs(m)}S_{obs(m)}^{(0)}\|_2 ,$$

*where $\nu_{obs(m)}$ is the smallest eigenvalue of $\Sigma_{obs(m)}$.*

Note that Proposition A.1 can easily be extended to the general case by working with $\Sigma/\rho(\Sigma)$ and multiplying the resulting approximation by $\rho(\Sigma)$, where $\rho(\Sigma)$ is the spectral radius of $\Sigma$.

*Proof.* Since the spectral radius of $\Sigma$ is strictly smaller than one, the spectral radius of each submatrix $\Sigma_{obs(m)}$ is also strictly smaller than one. This is a direct application of Cauchy Interlace Theorem [8] or it can be seen with the definition of the eigenvalues

$$\rho(\Sigma_{obs(m)}) = \max_{u \in \mathbb{R}^{|obs(m)|}} u^\top\Sigma_{obs(m)}u = \max_{\substack{x \in \mathbb{R}^d \\ x_{mis}=0}} x^\top\Sigma x \leq \max_{x \in \mathbb{R}^d} x^\top\Sigma x = \rho(\Sigma) .$$

Note that $S_{obs(m)}^\ell = \sum_{k=0}^{\ell-1}(Id - \Sigma_{obs})^k + (Id - \Sigma_{obs})^\ell S_{obs(m)}^0$ can be defined recursively via the iterative formula

$$S_{obs(m)}^\ell = (Id - \Sigma_{obs(m)})S_{obs(m)}^{\ell-1} + Id \qquad (38)$$

The matrix $(\Sigma_{obs(m)})^{-1}$ is a fixed point of the Neumann iterations (equation (38)). It verifies the following equation

$$(\Sigma_{obs(m)})^{-1} = (Id - \Sigma_{obs(m)})(\Sigma_{obs(m)})^{-1} + Id \ . \qquad (39)$$

By substracting 38 to this equation, we obtain

$$(\Sigma_{obs(m)})^{-1} - S_{obs(m)}^\ell = (Id - \Sigma_{obs(m)})((\Sigma_{obs(m)})^{-1} - S_{obs(m)}^{\ell-1}) \ . \qquad (40)$$

Multiplying both sides by $\Sigma_{obs(m)}$ yields

$$(Id - \Sigma_{obs(m)}S_{obs(m)}^\ell) = (Id - \Sigma_{obs(m)})(Id - \Sigma_{obs(m)}S_{obs(m)}^{\ell-1}) \ . \qquad (41)$$

Taking the $\ell_2$-norm and using Cauchy-Schwartz inequality yields

$$\|Id - \Sigma_{obs(m)}S_{obs(m)}^\ell\|_2 \le \|Id - \Sigma_{obs(m)}\|_2 \|Id - \Sigma_{obs(m)}S_{obs(m)}^{\ell-1}\|_2 \ . \qquad (42)$$

Let $\nu_{obs(m)}$ be the smallest eigenvalue of $\Sigma_{obs(m)}$, which is positive since $\Sigma$ is invertible. Since the largest eigenvalue of $\Sigma_{obs(m)}$ is upper bounded by 1, we get that $\|Id - \widetilde{\Sigma}\|_2 = (1 - \nu_{obs(m)})$ and by recursion we obtain

$$\|Id - \Sigma_{obs(m)}S_{obs(m)}^\ell\|_2 \le (1 - \nu_{obs(m)})^\ell \|Id - \Sigma_{obs(m)}S_{obs(m)}^0\|_2 \ . \qquad (43)$$

$\square$

## A.6  Proof of Proposition 3.1

**Proposition 3.1.** *Let $\nu$ be the smallest eigenvalue of $\Sigma$. Assume that the data are generated via a linear model defined in equation (1) and satisfy Assumption 1. Additionally, assume that either Assumption 2 or Assumption 3 holds and that the spectral radius of $\Sigma$ is strictly smaller than one. Then, for all $\ell \ge 1$,*

$$\mathbb{E}\left[(f_\ell^\star(X_{obs}, M) - f^\star(X_{obs}, M))^2\right] \le \frac{(1-\nu)^{2\ell}\|\beta^\star\|_2^2}{\nu}\mathbb{E}\left[\left\|Id - S_{obs(M)}^{(0)}\Sigma_{obs(M)}\right\|_2^2\right] \qquad (8)$$

According to Proposition 2.1 and the definition of the approximation of order $p$ of the Bayes predictor (see equations (7))

$$f_{\widetilde{X},\ell}^\star(\widetilde{X}) = \langle\beta_{obs}^\star, X_{obs}\rangle + \langle\beta_{mis}^\star, \mu_{mis} + \Sigma_{mis,obs}S_{obs}^{(\ell)}(X_{obs} - \mu_{obs})\rangle \ ,$$

Then

$$\mathbb{E}[(f_{\widetilde{X},\ell}^\star(\widetilde{X}) - f_{\widetilde{X}}^\star(\widetilde{X}))^2] \qquad (44)$$

$$= \mathbb{E}\left[\langle\beta_{mis}^\star \ , \ \Sigma_{mis,obs}(S_{obs}^\ell - \Sigma_{obs}^{-1})(X_{obs} - \mu_{obs})\rangle^2\right] \qquad (45)$$

$$= \mathbb{E}\left[(\beta_{mis}^\star)^\top\Sigma_{mis,obs}(S_{obs}^\ell - \Sigma_{obs}^{-1})(X_{obs} - \mu_{obs})(X_{obs} - \mu_{obs})^\top(S_{obs}^\ell - \Sigma_{obs}^{-1})\Sigma_{obs,mis}\beta_{mis}^\star\right] \qquad (46)$$

$$= \mathbb{E}\left[(\beta_{mis}^\star)^\top\Sigma_{mis,obs}(S_{obs}^\ell - \Sigma_{obs}^{-1})\underbrace{\mathbb{E}[(X_{obs} - \mu_{obs})(X_{obs} - \mu_{obs})^\top|M]}_{\Sigma_{obs}}(S_{obs}^\ell - \Sigma_{obs}^{-1})\Sigma_{obs,mis}\beta_{mis}^\star\right]$$
$$\qquad (47)$$

$$= \mathbb{E}\left[(\beta_{mis}^\star)^\top\Sigma_{mis,obs}(S_{obs}^\ell - \Sigma_{obs}^{-1})\Sigma_{obs}(S_{obs}^\ell - \Sigma_{obs}^{-1})\Sigma_{obs,mis}\beta_{mis}^\star\right] \qquad (48)$$

$$= \mathbb{E}\left[\left\|(\Sigma_{obs})^{\frac{1}{2}}(\Sigma_{obs})^{-1}(\Sigma_{obs}S_{obs}^\ell - Id_{obs})\Sigma_{obs,mis}\beta_{mis}^\star\right\|_2^2\right] \qquad (49)$$

$$= \mathbb{E}\left[\left\|(\Sigma_{obs})^{-\frac{1}{2}}(Id_{obs} - \Sigma_{obs}S_{obs}^\ell)\Sigma_{obs,mis}\beta_{mis}^\star\right\|_2^2\right] \qquad (50)$$

$$\le \|\Sigma^{-1}\|_2\|\Sigma\|_2^2\|\beta^\star\|_2^2\mathbb{E}\left[\|Id_{obs} - \Sigma_{obs}S_{obs}^\ell\|_2^2\right] \qquad (51)$$

$$\le \frac{1}{\nu}\|\beta^\star\|_2^2\mathbb{E}\left[(1 - \nu_{obs})^{2\ell}\|Id_{obs} - \Sigma_{obs}S_{obs}^0\|_2^2\right] \qquad (52)$$

An important point for going from (50) to (51) is to notice that for any missing pattern, we have

$$\|\Sigma_{obs,mis}\|_2 \le \|\Sigma\|_2 \text{ and } \|\Sigma_{obs}^{-1}\|_2 \le \|\Sigma^{-1}\|_2 \ .$$

The first inequality can be obtained by observing that computing the largest singular value of $\Sigma_{obs,mis}$ reduces to solving a constrained version of the maximization problem that defines the largest eigenvalue of $\Sigma$:

$$\|\Sigma_{obs,mis}\|_2 = \max_{\|x_{mis}\|_2=1} \|\Sigma_{obs,mis}x_{mis}\|_2 \leq \max_{\substack{\|x\|_2=1 \\ x_{obs}=0}} \|\Sigma_{obs,\cdot}x\|_2 \leq \max_{\substack{\|x\|_2=1 \\ x_{obs}=0}} \|\Sigma x\|_2 \leq \max_{\|x\|_2=1} \|\Sigma x\|_2^2 = \|\Sigma\|_2 \ .$$

where we used $\|\Sigma_{obs,\cdot}x\|_2^2 = \sum_{i \in obs}(\Sigma_i^\top x)^2 \leq \sum_{i=1}^d (\Sigma_i^\top x)^2 = \|\Sigma x\|_2^2$.

A similar observation can be done for computing the smallest eigenvalue of $\Sigma$, $\lambda_{\min}(\Sigma)$:

$$\lambda_{\min}(\Sigma) = \min_{\|x\|_2=1} x^\top \Sigma x \leq \min_{\substack{\|x\|_2=1 \\ x_{mis}=0}} x^\top \Sigma x = \min_{\|x_{obs}\|_2=1} x_{obs}^\top \Sigma_{obs} x_{obs} = \lambda_{\min}(\Sigma_{obs}) \ .$$

and we can deduce the second inequality by noting that $\lambda_{\min}(\Sigma) = \frac{1}{\|\Sigma^{-1}\|_2^2}$ and $\lambda_{\min}(\Sigma_{obs}) = \frac{1}{\|\Sigma_{obs}^{-1}\|_2^2}$.

### A.7 Proof of Proposition 3.2

**Proposition 3.2** (equivalence MLP - depth-1 NeuMiss network). *Let $[X \odot (1-M), M] \in [0,1]^d \times \{0,1\}^d$ be an input $X$ imputed by 0 concatenated with the mask $M$.*

- *Let $\mathcal{H}_{ReLU} = \left(W \in \mathbb{R}^{d \times 2d}, ReLU\right)$ be a hidden layer which connects $[X \odot (1-M), M]$ to $d$ hidden units, and applies a ReLU nonlinearity to the activations.*
- *Let $\mathcal{H}_{\odot M} = \left(W \in \mathbb{R}^{d \times d}, \mu, \odot M\right)$ be a hidden layer that connects an input $(X - \mu) \odot (1-M)$ to $d$ hidden units, and applies a $\odot M$ nonlinearity.*

*Denote by $h_k^{ReLU}$ and $h_k^{\odot M}$ the outputs of the $k^{th}$ hidden unit of each layer. Then there exists a configuration of the weights of the hidden layer $\mathcal{H}_{ReLU}$ such that $\mathcal{H}_{\odot M}$ and $\mathcal{H}_{ReLU}$ have the same hidden units activated for any $(X_{obs}, M)$, and activated hidden units are such that $h_k^{ReLU}(X_{obs}, M) = h_k^{\odot M}(X_{obs}, M) + c_k$ where $c_k \in \mathbb{R}$.*

**Obtaining a $\odot M$ nonlinearity from a ReLU nonlinearity.** Let $\mathcal{H}_{ReLU} = \left(\left[W^{(X)}, W^{(M)}\right] \in \mathbb{R}^{d \times 2d}, ReLU\right)$ be a hidden layer which connects $[X, M]$ to $d$ hidden units, and applies a ReLU nonlinearity to the activations. We denote by $b \in \mathbb{R}^d$ the bias corresponding to this layer. Let $k \in [\![1, d]\!]$. Depending on the missing data pattern that is given as input, the $k^{th}$ entry can correspond to either a missing or an observed entry. We now write the activation of the $k^{th}$ hidden unit depending on whether entry $k$ is observed or missing. The activation of the $k^{th}$ hidden unit is given by

$$a_k = W_{k,\cdot}^{(X)} X + W_{k,\cdot}^{(M)} M + b_k \tag{53}$$

$$= W_{k,obs}^{(X)} X_{obs} + W_{k,mis}^{(M)} \mathbf{1}_{mis} + b_k. \tag{54}$$

Emphasizing the role of $W_{k,k}^{(M)}$ and $W_{k,k}^{(X)}$, we can decompose equation (54) depending on whether the $k^{th}$ entry is observed or missing

$$\text{If } k \in mis, \quad a_k = W_{k,obs}^{(X)} X_{obs} + W_{k,k}^{(M)} + W_{k,mis\setminus\{k\}}^{(M)} \mathbf{1}_{k,mis\setminus\{k\}} + b_k \tag{55}$$

$$\text{If } k \in obs, \quad a_k = W_{k,k}^{(X)} X_k + W_{k,obs\setminus\{k\}}^{(X)} X_{obs\setminus\{k\}} + W_{k,mis}^{(M)} \mathbf{1}_{mis} + b_k. \tag{56}$$

Suppose that the weights $W^{(X)}$ as well as $W_{i,j}^{(M)}, i \neq j$ are fixed. Then, under the assumption that the support of $X$ is finite, there exists a bias $b_k^*$ which verifies:

$$\forall X, \quad a_k = W_{k,k}^{(X)} X_k + W_{k,obs\setminus\{k\}}^{(X)} X_{obs\setminus\{k\}} + W_{k,mis}^{(M)} \mathbf{1}_{mis} + b_k^* \leq 0 \tag{57}$$

i.e., there exists a bias $b_k^*$ such that the activation of the $k^{th}$ hidden unit is always negative when $k$ is observed. Similarly, there exists $W_{k,k}^{*,(M)}$ such that:

$$\forall X, \quad a_k = W_{k,obs}^{(X)} X_{obs} + W_{k,k}^{*,(M)} + W_{k,mis\setminus\{k\}}^{(M)} \mathbf{1}_{k,mis\setminus\{k\}} + b_k^* \geq 0 \tag{58}$$

i.e., there exists a weight $W_{k,k}^{*,(M)}$ such that the activation of the $k^{th}$ hidden unit is always positive when $k$ is missing. Note that these results hold because the weight $W_{k,k}^{(M)}$ only appears in the expression of $a_k$ when entry $k$ is missing. Let $h_k = ReLU(a_k)$. By choosing $b_k = b_k^*$ and $W_{k,k}^{(M)} = W_{k,k}^{*,(M)}$, we have that:

$$\text{If } k \in mis, \quad h_k = a_k \tag{59}$$

$$\text{If } k \in obs, \quad h_k = 0 \tag{60}$$

As a result, the output of the hidden layer $\mathcal{H}_{ReLU}$ can be rewritten as:

$$h_k = a_k \odot M \tag{61}$$

i.e., a $\odot M$ nonlinearity is applied to the activations.

**Equating the slopes and biases of $\mathcal{H}_{ReLU}$ and $\mathcal{H}_{\odot M}$.** Let $\mathcal{H}_{\odot M} = \left( W \in \mathbb{R}^{d \times d}, \mu, \odot M \right)$ be the layer that connect $(X - \mu) \odot (1 - M)$ to $d$ hidden units via the weight matrix $W$, and applies a $\odot M$ nonlinearity to the activations. We will denote by $c \in \mathbb{R}^d$ the bias corresponding to this layer.

The activations for this layer are given by:

$$a_k = W_{k,obs}(X_{obs} - \mu_{obs}) + c_k \tag{62}$$
$$= W_{k,obs}X_{obs} - W_{k,obs}\mu_{obs} + c_k \tag{63}$$

Then by applying the non-linearity we obtain the output of the hidden layer:

$$\text{If } k \in mis, \quad h_k = a_k \tag{64}$$
$$\text{If } k \in obs, \quad h_k = 0 \tag{65}$$

It is straigthforward to see that with the choice of $b_k = b_k^*$ and $W_{k,k}^{(M)} = W_{k,k}^{*,(M)}$ for $\mathcal{H}_{ReLU}$, both hidden layers have the same output $h_k = 0$ when entry $k$ is observed. It remains to be shown that there exists a configuration of the weights of $\mathcal{H}_{ReLU}$ such that the activations $a_k$ when entry $k$ is missing are equal to those of $\mathcal{H}_{\odot M}$. To avoid confusions, we will now denote by $a_k^{(N)}$ the activations of $\mathcal{H}_{\odot M}$ and by $a_k^{(R)}$ the activations of $\mathcal{H}_{ReLU}$. We recall here the activations for both layers as derived in 63 and 55.

$$\text{If } k \in mis, \begin{cases} a_k^{(N)} = W_{k,obs}X_{obs} - W_{k,obs}\mu_{obs} + c_k \\ a_k^{(R)} = W_{k,obs}^{(X)}X_{obs} + W_{k,k}^{*,(M)} + W_{k,mis\setminus\{k\}}^{(M)}\mathbf{1}_{k,mis\setminus\{k\}} + b_k^* \end{cases} \tag{66}$$

By setting $W_{k,.}^{(X)} = W_{k,.}$, we obtain that both activations have the same slopes with regards to $X$. We now turn to the biases. We have that:

$$W_{k,k}^{*,(M)} + W_{k,mis\setminus\{k\}}^{(M)}\mathbf{1}_{k,mis\setminus\{k\}} + b_k^* = W_{k,.}^{(M)}\mathbf{1} - W_{k,obs}^{(M)}\mathbf{1} + b_k^* \tag{67}$$

We now set:

$$\forall j \in obs, \quad W_{kj}^{(M)} = W_{kj}\mu_j \tag{68}$$
$$W_{k.}^{(M)}\mathbf{1} + b_k^* = c_k \tag{69}$$

to obtain that both activations have the same biases. Note that 68 sets the weights $W_{k,j}$ for all $j \neq k$ (since $obs$ can contain any entries except $k$). As a consequence, equation 69 implies an equation invloving $W_{kk}^{*,(M)}$ and $b_k^*$ where all other parameters have already been set. Since $W_{kk}^{*,(M)}$ and $b_k^*$ are also chosen to satisfy the inequalities 57 and 58, it may not be possible to choose them so as to also satify equation 69. As a result, the functions computed by the activated hidden units of $\mathcal{H}_{ReLU}$ can be equal to those computed by $\mathcal{H}_{\odot M}$ up to a constant.

# B   Additional results

## B.1   NeuMiss network scaling law in MNAR

## B.2   NeuMiss network performances in MAR

The MAR data was generated as follows: first, a subset of variables with *no* missing values is randomly selected (10%). The remaining variables have missing values according to a logistic model with random weights, but whose intercept is chosen so as to attain the desired proportion of missing values on those variables (50%). As can be seen from figure 6, the trends observed for MAR are the same as those for MCAR.

Figure 5: **Required capacity in various MNAR settings** — Top: Gaussian self-masking, bottom: probit self-masking. Performance of NeuMiss networks varying the depth in simulations with different number of samples $n$ and of features $d$.

Figure 6: **Predictive performances in MAR scenario** — varying number of samples $n$, and number of features $d$. All experiments are repeated 20 times.