[Reviews · NeurIPS 2020]

Review 1

Summary and Contributions: The paper addresses linear regression problem in the presence of missing data that may be Missing Not At Random. It first derives the analytical form of the Bayes predictor under Missing at Random (MAR) and Gaussian self-masking missing data mechanisms, then proposes a neural network architecture to approximate the Bayes predictor. The proposed learning method is empirically shown to perform well.

Strengths: The paper addresses a relevant problem, the proposed method is novel, and the results should be useful in practice.

Weaknesses: The proposed method is limited to learning linear models.

Correctness: The claims look like correct.

Clarity: The paper is well written.

Relation to Prior Work: The relation to prior work is clearly discussed.

Reproducibility: Yes

Additional Feedback: What is v in Equation (8)? ===After author rebuttal === My opinion has not changed. I think the paper should be accepted.


Review 2

Summary and Contributions: The authors derive the optimal linear predictor under various missing data mechanisms including missing at random (MAR) and self-masking missing not at random (MNAR). Specifically, a new architecture, called Neumann networks, is presented based on Neumann series approximation of the Bayes predictors. Many methods associated with missing data have been proposed so far, but the proposed method is significant in the sense that it can handle missing not at random mechanism. Moreover, the proposed method scales well unlike the conventional methods for MNAR model. In the experiment, it is shown that the proposed method is more robust than the conventional methods especially in MNAR case.

Strengths: The derivation of the expression of the Bayes predictor under the various missing value mechanisms is novel and theoretically valid. Furthermore, Neuman network's approximation error bound is also theoretically derived. I think this work is an important achievement in the research of missing data analysis.

Weaknesses: In the derivation data X is assumed to be Gaussian, and the predictor is assumed to be linear model. Due to these assumptions, there seems to be application limits.

Correctness: The authors argue that the proposed method is more robust than other methods for all missing mechanisms, but in fact, as shown in Fig. 4, when the number of samples is small, the conventional method has better performance in MCAR. Smaller numbers of data should be compared even in the MNAR case.

Clarity: Introduction is great. The paper is well written and formal.

Relation to Prior Work: The authors reviews the conventional methods for various missing data mechanisms. Moreover, the difference between the proposed method and the others is clear.

Reproducibility: Yes

Additional Feedback: Typo: Minus sign is missing of R2-Bayes rate (d=50 case) for MCAR in Fig.4.


Review 3

Summary and Contributions: The paper derives analytical expressions of optimal predictors in the presence of Missing Completely At Random (MCAR), Missing At Random (MAR) and self-masking missingness in the linear Gaussian case. Then, the paper proposes Neumann Network for learning the optimal predictor in the MAR case and show the insights and connection to the neural network with ReLU activations. There are two challenges of learning the optimal predicator from data containing missing values: 1) computing the inversion of covariance matrices in the MAR optimal predicator; 2) 2^d optimal predictors with different missingness patterns required to learn the optimal predictor, where d is the number of features/covariates. For the first one, the paper provides a theoretical analysis, which is approximated in a recursive manner with the convergence and upper bounder guarantee. For the second one, the Neumann Network shares the weights of optimal predictors with different missing patterns, which turns out empirically more data efficient and robust to self-masking missingness cases.

Strengths: The analytical expression and discussion about the optimal predictor in self-masking missingness in Section 2 is novel and significant in the study of missing data problems. This could be helpful for understanding and dealing with the self-masking missingness, which is still a challenge in many fields. The theoretical analysis guided neural network is an interesting and neat way to solve the learning problem. The neural network is guided by an approximation method with the analysis of convergence and upper bound. Moreover, the theoretical and empirical analysis of the neural network is appreciated very much, which shows the connection to the ReLU network and required number of samples.

Weaknesses: The approximation and Neumann network are based on the expression of the optimal predictor in MAR, which is a less interesting problem, even though it shows that the performance of the proposed method works in self-masking missingness cases empirically. The lines 190-192 said, there could be another similar network for self-masking missingness, which maybe less satisfying. I am not asking for a solution/implementation for self-masking missingness cases here, but I would like to see discussion or proper justification for why and when the proposed method can be used for the self-masking missingness ( even for MNAR, the larger class of missingness mechanisms ) under which kind of guarantee. Then the contribution for the self-masking missingness ( even MNAR ) problem would be much more helpful for the community. ===Update === My opinion has not changed. The equation (2) in author feedback is based on the approximation which is claimed to be "poor" in the paper.

Correctness: The comments are about the experiments. The paper claimed that the method is robust to the MNAR case, which could be not suitable. Note that the MNAR is a large class of missingness mechanisms, which is much larger than M(C)AR together with self-masking missingness. In fact, the mentioned “MNAR” in the paper ( as in lines 77, 246, and 289-290 ) would be more accurate if replaced with self-masking missingness because the analysis, discussion and experiments are all based on the self-masking missingness. I would like to know why the experiments didn’t test on the MAR scenario, which is quite different with MCAR in many scenarios. Moreover, I am curious how the method would perform in the case where MAR and self-masking missingness happen at the same time.

Clarity: Yes, it is well written.

Relation to Prior Work: Yes.

Reproducibility: Yes

Additional Feedback: Some tiny things: > In the lines 46-48, I am not sure that I understand it. > Line 106, maybe “= P(M = m)”. ===Update === My opinion has not changed. Although the assumptions may make the work limited, I think the paper should be accepted because of the theoretical analysis and the neural network design.


Review 4

Summary and Contributions: The authors of this paper propose a method for supervised learning with missing covariates. They establish the Bayes predictor for a variety of specifying missing data mechanisms including MNAR mechanisms. They suggest a particular neural network architecture for learning the Bayes predictor and demonstrate it's performance empirically.

Strengths: I thought this was a really nice paper. They take a different approach / perspective to missing data than I commonly encounter. I think this is a useful contribution in for very important practical problem that is often not given rigorous treatment. There are some nice insights about the Bayes predictor in the simple linear setting which motivates a particular network architecture. Specifically, I think this is a nice first step toward theory and best practice for dealing with missing data in neural network models. Although I didn't follow every detail, I thought the section comparing their network to the MLP fit with the concatenated data to be particularly interesting, as it links common practice to some of the theory the authors are developing. Update: I really enjoyed this paper and think it should be accepted. The authors responses don't change my view in this regard. I'm keeping my score.

Weaknesses: The required assumptions for the key propositions are quite strong (Gaussian data, Gaussian self-masking etc). How robust are the networks to deviations for Gaussian data? It would be nice to see some robustness checks on non-Gaussian data.

Correctness: To the best of my knowledge, the method appears correct.

Clarity: The paper is very well written and quite clear. I couldn't really make sense of "Differences between MNAR and M(C)AR predictors" Section. It seems like the authors have some useful insights to share but perhaps given the length limit, I wasn't quite able to identify the main point of this Section.

Relation to Prior Work: The prior work is clearly discussed. That said, one difficulty is that this references and builds on very recent work (papers on arxiv 2019/2020) which personally I was not familiar. Given the recency of the works they reference, it may help to add a few more sentences about the main contributions in the previous work and the extensions in this work.

Reproducibility: Yes

Additional Feedback:

[Author Response · NeurIPS 2020]

We thank the reviewers for their time and relevant comments.

**Why we use our architecture to approximate the self-masking Bayes predictor - to rev. #3 and #4:** Reviewer #4 mentioned a lack of clarity in the paragraph entitled "Differences between MNAR and M(C)AR predictors" (l.128-133). This paragraph actually touches upon a point raised by Reviewer #3: why our method can be used for self-masking missingness (also discussed l.190-192). These questions suggest that this paragraph should be more detailed. Hereafter, we explain it in more details and give an answer to reviewer #3.

The expression of the M(C)AR Bayes predictors is given by (eq. 4 in the paper):

$$f^\star(X_{obs}, M) = \beta_0^\star + \langle \beta_{obs}^\star, X_{obs} \rangle + \langle \beta_{mis}^\star, \mu_{mis} + \Sigma_{mis,obs}(\Sigma_{obs})^{-1}(X_{obs} - \mu_{obs}) \rangle \tag{1}$$

The expression of the (MNAR) self-masking Bayes predictor is more complicated (eq. 5 in the paper). To study this expression, we approximate $D_{mis}\Sigma_{mis|obs}^{-1}$ by $Id$. Then, the self-masking Bayes predictor becomes:

$$f^\star(X_{obs}, M) \approx \beta_0^\star + \langle \beta_{obs}^\star, X_{obs} \rangle + \langle \beta_{mis}^\star, \frac{1}{2}(\tilde{\mu}_{mis} + \mu_{mis}) + \frac{1}{2}\Sigma_{mis,obs}(\Sigma_{obs})^{-1}(X_{obs} - \mu_{obs}) \rangle \tag{2}$$

Thus, under this approximation, the self-masking Bayes predictor can be modeled by our proposed architecture (just as the M(C)AR Bayes predictor), the only difference being the targeted value for parameter $\mu$ of the network (in blue in the two models above) and a scaling factor of $1/2$ for $W_{mix}$ (in orange). A less coarse approximation also works: $D_{mis}\Sigma_{mis|obs}^{-1} \approx \hat{D}_{mis}$ where $\hat{D}$ is a diagonal matrix. In this case, the proposed architecture can perfectly model the self-masking Bayes predictor: the parameter $\mu$ of the network should target $(Id + \hat{D})^{-1}(\tilde{\mu} + \hat{D}\mu)$ and $W_{mix}$ should target $(Id + \hat{D})^{-1}\hat{D}\Sigma$ instead of simply $\Sigma$ in the M(C)AR case. Consequently, **our architecture can well approximate the self-masking Bayes predictor by adjusting the values learned for the parameters $\mu$ and $W_{mix}$ if $D_{mis}\Sigma_{mis|obs}^{-1}$ are close to diagonal matrices**. We will add this discussion to the Appendix.

**Experimental results under the MAR scenario - to rev.#3** The results are presented in the figure below:

The MAR data was generated as follows: first, a subset of variables with *no* missing values is randomly selected (10%). The remaining variables have missing values according to a logistic model with random weights, re-scaled so as to attain the desired proportion of missing values on those variables (50%). Note that we cannot compute the Bayes rate, so instead of showing (R2 - Bayes rate) we show (R2 - best R2).

As can be seen from the figure, the trends oberved for MAR are the same as those for MCAR. We will add this figure to the appendix. We have not tested the scenario where MAR and self-masking missingness happen at the same time.

**Modeling non linear functions - to rev. #1 and #2** The reviewers pointed out that the proposed architecture is limited to linear models. Indeed, our theoretical foundations derive from the study of linear models. However, as a differentiable architecture, it can be readily included as a building block in more complex networks. For example, the layer $W_\beta$ can be replaced by a MLP.

**Robustness to non Gaussian data - to rev. #2 and #3** The crux of the method is to capture the covariance of the data, which relates the different slopes of the models on incomplete data. This covariance will be relevant even on non-Gaussian data, though we can so far only develop formal arguments under Gaussian assumptions.

**What is $\nu$ in eq.(8) - to rev. #1** It is the smallest eigenvalue of the covariance matrix $\Sigma$. We apologize, the definition was inadequately moved to the appendix; we will move it to the main text.

**What happens with few samples - to rev. #2** Reviewer #2 made a comment related to the number of samples. The difficulty of a problem is linked to the ratio # dimensions/# samples, the higher the ratio, the more difficult the problem. The highest ratio for which we presented experiments is for dimension 50 and 20000 samples. For such ratio and higher, using a depth of 0 for our architecture is enough, more depth triggers overfitting. We showed theoretically in section 3.4 that the MLP can model a depth-0 Neumann network. Thus the MLP, whether in MCAR or selfmaking, is on par or slightly better than the Neumann network for high ratios. As for MICE+LR, it has an advantage over MLP and Neumann for high ratios in MCAR, but not in selfmasking, because Mice assumes the data to be M(C)AR.

[Meta-Review · NeurIPS 2020]

The paper attacks the classical problem of linear regression with missing values. It computes the Bayes predictor in several cases with missing values and then uses Neumann series to approximate the Bayes predictor. This approximation is then used to design Neural Networks with RelU functions. The propositions describing self-masking missingness, appears to be a novel concept, are interesting but can be considered slightly restrictive because of Linear Gaussian assumptions. However, both the results and the methods should be of interest to NeuriPS 2020 community.